

# Improving prediction of maternal health risks using PCA features and TreeNet model

Leila Jamel[1], Muhammad Umer[2], Oumaima Saidani[1], Bayan Alabduallah[1], Shtwai Alsubai[3], Farruh Ishmanov[4], Tai-hoon Kim[5] and Imran Ashraf[6]

[1] Department of Information Systems, College of Computer and Information Sciences, Princess Nourah bint Abdulrahman University, Riyadh, Saudi Arabia
[2] Department of Computer Science & Information Technology, The Islamia University of Bahawalpur, Bahawalpur, Punjab, Pakistan
[3] Department of Computer Science, College of Computer Engineering and Sciences, Prince Sattam Bin Abdulaziz University, Al-Kharj, Saudi Arabia
[4] Department of Electronics and Communication Engineering, Kwangwoon University, Seoul, Republic of South Korea
[5] School of Electrical and Computer Engineering, Yeosu Campus, Chonnam National University, Daehak-ro, Yeosu-si, Jeollanam-do, Republic of South Korea
[6] Department of Information and Communication Engineering, Yeungnam University, Gyeongsan, Republic of South Korea

Corresponding authors
Tai-hoon Kim,
taihoonn@chonnam.ac.kr
Imran Ashraf, imranashraf@ynu.ac.kr

## ABSTRACT

Maternal healthcare is a critical aspect of public health that focuses on the well-being of pregnant women before, during, and after childbirth. It encompasses a range of services aimed at ensuring the optimal health of both the mother and the developing fetus. During pregnancy and in the postpartum period, the mother's health is susceptible to several complications and risks, and timely detection of such risks can play a vital role in women's safety. This study proposes an approach to predict risks associated with maternal health. The first step of the approach involves utilizing principal component analysis (PCA) to extract significant features from the dataset. Following that, this study employs a stacked ensemble voting classifier which combines one machine learning and one deep learning model to achieve high performance. The performance of the proposed approach is compared to six machine learning algorithms and one deep learning algorithm. Two scenarios are considered for the experiments: one utilizing all features and the other using PCA features. By utilizing PCA-based features, the proposed model achieves an accuracy of 98.25%, precision of 99.17%, recall of 99.16%, and an F1 score of 99.16%. The effectiveness of the proposed model is further confirmed by comparing it to existing state of-the-art approaches.

## INTRODUCTION

The maternal mortality ratio indicates the number of women who die from pregnancy-related complications per 100,000 live births (*WHO, 2019*). The data provided by the World Health Organization (WHO) indicate that an average of 808 women lost their

lives each day in 2017 due to complications related to pregnancy (*Roser & Ritchie, 2021*). When analyzing maternal deaths on a global scale, it becomes evident that approximately two-thirds (200,000) of these fatalities occurred in sub-Saharan Africa, while only 19% (57,000) took place in South Asia (*Mehboob et al., 2021*). In the year 2017, five countries with the highest number of maternal deaths were Tanzania(11,000), Ethiopia (14,000), the Democratic Republic of the Congo (16,000), India (35,000), and Nigeria (67,000) (*Raza et al., 2022*).

Pregnant women's health is influenced by several factors including age and blood disorders such as high or low blood pressure, blood glucose levels, body temperature, and heart rate. These factors directly increase the risk of complications during pregnancy which can lead to the unfortunate loss of both the woman's pregnancy and her life. Addressing these health factors through specialized medical interventions is crucial. Early prediction of these risks can potentially empower medical experts to take timely and appropriate actions to reduce the likelihood of maternal mortality.

The risk of complications during pregnancy, which can result in both the loss of the pregnancy and the woman's life can be directly influenced by factors such as age and blood disorders. It is essential to address these health issues through specialized medical interventions since the early identification of such hazards may allow medical professionals to take the necessary steps to lower the possibility of maternal mortality. Pregnancy-related medical issues and mortality which affect both mothers and their newborns' health are currently a major global concern. Around 287,000 women passed away in pregnancy and childbirth in the year 2020 (*WHO, 2023*). The substantial differences in global access to medical care and treatment are highlighted by the uneven distribution of mortality. Furthermore, there are considerable differences in mortality rates not just across countries but also within them, which have an impact on both high and low-income women as well as those living in urban and rural areas. Pregnancy and delivery problems continue to be significant causes of mortality in underdeveloped nations (*Abubakar, Tillmann & Banerjee, 2015*).

Even though most of these issues begin during pregnancy, others may develop beforehand and become worse throughout it. It is significant to underline that almost all of these maternal deaths take place in settings with a shortage of resources and the majority of them might have been avoided or treated with proper funding and care. Preeclampsia, infections, gestational diabetes, hypertension, pregnancy loss, miscarriage, premature labor, and stillbirth are a few of the most typical pregnancy issues. Severe nausea, vomiting, and anemia brought on by a lack of iron are further potential issues (*National Institute of Health (NIH), 2021*; *Grivell et al., 2015*). As a result, these disorders can greatly raise the risks to the growth of a pregnancy demanding the development of novel ways for monitoring and evaluating the fetus's health. In recent years, artificial intelligence (AI) has been used in a variety of fields to solve a variety of issues (*Kaur et al., 2020*). These AI-based approaches assist in understanding and learning complex correlations between factors. Machine learning approaches can produce extremely precise results especially when working with massive amounts of input data (*Manifold et al., 2021*).

With the use of various types of data such as images, electronic health records (EHRs), and time-series data, machine learning-based models have been widely used in the medical field to handle a wide range of tasks including disease prediction. These models can find patterns in medical data that were previously unknown enabling health professionals to make quick and precise diagnoses (*Zeng et al., 2019*). Machine learning can be used to undertake highly accurate analyses of various infections enabling health professionals to offer better treatment options. As a result, machine learning aids in making better medical decisions. Additionally, machine learning helps doctors by assisting with patient care (*Berrar & Dubitzky, 2021*). Machine learning algorithms improve the accuracy of diagnoses by examining both organized and unstructured medical records including diagnosis data (*Theis et al., 2021*). Medical imaging, healthcare data analytics, maternal health care, breast cancer identification, heart disease analysis, and diabetes detection are just a few of the many fields where machine learning is being used in medicine. Focusing on the potential of machine learning, this study designs an approach for predicting pregnancy-related health risks and makes the following contributions

- This study introduces an ensemble model that aims to predict maternal health during pregnancy. The proposed ensemble model combines an extra tree classifier (ETC) and a multi-layer perceptron (MLP), utilizing a voting mechanism to generate the final prediction.
- The present study employs principle component analysis (PCA) to extract significant features from the dataset, which directly contribute to the prediction of maternal health during pregnancy.
- A comparative analysis of performance is conducted using multiple machine learning models including logistic regression (LR), extreme gradient boosting (XGBoost), random forest (RF), decision tree (DT), ETC, and stochastic gradient descent (SGD).
- This research work also makes use of two deep learning models MLP and convolutional neural network (CNN) for performance comparison. Furthermore, the proposed model's effectiveness is analyzed by comparing its performance to state-of-the-art approaches in terms of accuracy, precision, recall, and F1 score.

The following structure is used for the remaining sections of this study: 'Related Work' covers the review of related research. 'Material and Methods' outlines the dataset, proposed approach, evaluation parameters, and the machine learning models employed for predicting maternal health. 'Results and Discussion' presents the experimental setup, results obtained using each learning model, discussion, comparison, and explanation of each result using the XAI technique. Finally, 'Conclusions' concludes the study and suggests potential avenues for future research.

## RELATED WORK

Several researchers have developed models to predict health risks during pregnancy as a result of their recognition of the importance of maternal health. These techniques incorporate both conventional and machine learning methods. Some research works concentrate on identifying and documenting the health risk factors present in pregnant

women whereas others concentrate on anticipating these risks. Risk analysis, risk prediction, and the use of AI-based techniques for disease diagnosis are currently popular trends. Continuous research observations are made over time at the Daffodil International University in Dhaka, Bangladesh. These observations focus on several health risk variables, such as age (below 20 or above 35), past birth experience, history of pregnancy problems, and miscarriage.

During the pregnancy, *Özsezer & Mermer (2021)* worked on the health risk analysis. Data processing, hyperparameter tuning, modeling, and comparative analysis are the four divisions of the work. To predict health risks during pregnancy, the authors used eleven machine learning models including k-nearest neighbor (KNN), XGBoost, Light gradient boosting machine (GBM), artificial neural network (ANN), LR, CatBoost, RF, support vector machine (SVM), and classification and regression tree (CART). The results show that the LightGBM and CatBoost exhibit the highest accuracy of 88%. On the risk prediction for maternal health, *Raza et al. (2022)* proposed an ensemble learning-based feature engineering method for the effective analysis of maternal health data. The authors focused on creating an AI-based system for predicting risks to maternal health. With the DT-BiLTCN feature extraction technique, they used a variety of machine learning models. Experimental results indicate that the SVM with ensemble features achieves a 98% accuracy.

*Ramla, Sangeetha & Nickolas (2018)* proposed an effective approach to lower the rate of maternal and fetal death by analyzing the data related to pregnancy. They introduced the CART binary decision tree to predict high pregnancy risk. The cardiotocography dataset from UCI which included 2,126 fatal cardiotocographs was used for experiments. Using a 5-fold cross-validation, the model produces a good accuracy of 88%. Similarly, to predict the maternal risk level, *Raza et al. (2022)* employed a range of machine learning models including the k-NN, Naive Bayes (NB), neural network (NN), RF, and stack models. The data is divided into high, low, and medium maternal health risk classes. The study's results indicate that RF shows better results with an accuracy value of 83%. *Irfan, Basuki & Azhar (2021)* proposed an interpretable machine learning method for the automatic prediction of maternal health risk. The authors deployed the model with several feature selection techniques. Results indicate that with an accuracy score of 94%, the XGBoost model outperformed other learning models. To obtain insights, the authors used the LIME and SHAP interpretability for the classification. *Alam, Patwary & Hassan (2021)* proposed a bagging ensemble model for the prediction of birth mode in Bangladeshi women. k-NN, DT, and SVM are implemented separately, as well as, with the bagging ensemble. The findings show that bagging ensemble models outperformed the traditional models. Additionally, the authors demonstrated a link between important variables and the prevalence of cesarean procedures.

*Pawar et al. (2022)* used a machine learning-based approach for the risk prediction of maternal health. Traditional machine learning models like DT, NB, MLP, J48, LMT, RF, REP tree, and bagging were employed. The findings demonstrate that the RF model has an accuracy of 70.21% which is superior to other models used in the study. Similarly, a machine learning-based approach was proposed by *Assaduzzaman, Al Mamun & Hasan (2023)* for the early prediction of maternal health risk. To effectively address the abnormalities in the

**Table 1  Summary of the related work.**

| Ref | Classifiers | Dataset |
| --- | --- | --- |
| *Özsezer & Mermer (2021)* | KNN, XGBoost, Light GBM, ANN, LR, CatBoost, RF, SVM, GBM, and CART | Kaggle |
| *Raza et al. (2022)* | DTC, LR, KNN, ETC, RFC, SVM | Kaggle |
| *Ramla, Sangeetha & Nickolas (2018)* | CART, DT | UCI |
| *Raza et al. (2022)* | k-NN, NB, NN, RF, and stacked Generalization | UCI |
| *Irfan, Basuki & Azhar (2021)* | RF, NB, KNN, XGBoost With three feature selection methods (CFS, C5.0, KSPR) | Cipto Mulyo Malang Public Health Cente, dataset |
| *Alam, Patwary & Hassan (2021)* | NB, NB (Bagging), k-NN, k-NN (Bagging), DT, DT (Bagging), SVM and SVM (Bagging) | BDHS-2014 dataset |
| *Pawar et al. (2022)* | DT, NB, MLP, J48, LMT, RF, REP tree, Bagging | UCI (s) |
| *Assaduzzaman, Al Mamun & Hasan (2023)* | RF, DT, CatBoost, GBC, XGBoost | UCI (1014) |
| *Ahmed & Kashem (2020)* | DT, RF, SVM, Sequential Minimal Optimization, NB, LR, Logistic model tree | IoT sensor dataset (self-collected) |
| *Marques et al. (2020)* | KNN, SVM , RF, and 1D-CNN | IoT sensor data |

data value, they applied a variety of feature engineering and data pre-processing techniques. The study's results reveal that the RF has the highest accuracy value.

*Ahmed & Kashem (2020)* proposed an Internet of Things (IoT)-based system for the early prediction of maternal health. The authors collected the data from several hospitals in Bangladesh using IoT-based sensors. The results of using different machine learning algorithms on the data from wearable sensors show that the modified DT attained a maximum accuracy of 98.51%. For high-risk pregnancies, *Marques et al. (2020)* proposed a comprehensive system for monitoring maternal and fetal signals. Their approach involves utilizing IoT sensors to collect data, extracting relevant features using data analytics techniques, and incorporating an intelligent diagnostic aid system that employs a 1-D CNN classifier. The results showed that 1D-CNN achieved the highest accuracy of 92.51%.

The research reviewed offers a comprehensive overview of diverse approaches and models used in predicting maternal health risks during pregnancy. Researchers have employed an array of machine learning techniques and results among these models suggest both the complexity and potential effectiveness of leveraging AI techniques in this domain. These studies not only highlight the significance of predictive models but also emphasize the potential for further exploration and refinement in this critical domain of healthcare. Given the importance of various machine learning and deep learning models discussed in existing literature, a critical summary is provided in Table 1.

**Table 2  Description of maternal health dataset.**

| Attribute | Description | Data type | Range/Values |
|---|---|---|---|
| Age | It represents the age of a woman when she is pregnant | Numerical | 10–70 years |
| Bs | It shows the blood glucose level in mmol/L during the pregnancy. | Numerical | 6–19 mmol/L |
| RiskLevel | It represents the intensity of the risk during pregnancy. | Categorical | High, Low, Mild |
| SystolicBP | It represents the higher or upper value of blood pressure during pregnancy, measured in millimeters of mercury (mmHg). | Numerical | 70–160 mmHg |
| HeartRate | It represents the heart rate measured in beats per minute (BPM). | Numerical | 7–90 BPM |
| DiastolicBP | It represents the lower or bottom value of blood pressure during pregnancy, measured in millimeters of mercury (mmHg). | Numerical | 49–100 mmHg |
| Bodytemp | It shows the body temperature of the pregnant women | Numerical | 90–103 Fahrenheit |

# MATERIALS AND METHODS

This particular section of the study provides a concise summary of the dataset utilized for predicting maternal health risk. It also encompasses an explanation of the PCA feature engineering technique, a depiction of the machine learning and deep learning models employed, and an introduction to the proposed tree model.

## Dataset for experiments

The dataset used in this study was originally created by Marzia et al. from Daffodil International University in Dhaka, Bangladesh, and is publicly available (*UCI Machine Learning Repository, 2021*). The dataset has been collected using an IoT-based risk monitoring system implemented in various healthcare facilities, including hospitals, community clinics, and maternal health centers (*Afreen & Bajwa, 2021*). Additionally, a benchmark dataset with similar characteristics is available on *Kaggle (2022)*. The dataset consists of seven features: Age, Bs, RiskLevel, SystolicBP, HeartRate, DiastolicBP, and Bodytemp, which are used as target classes. A comprehensive description of the maternal health dataset including details of its attributes is provided in Table 2. The dataset contains a total of 1014 samples of maternal health data, with 406 instances classified as low-risk, 336 as mid-risk, and 272 as high-risk.

## Data preprocessing and feature selection

The data preprocessing includes the label encoding of the categorical attribute 'RiskLevel'. The second step of preprocessing includes the feature selection technique employed to identify the most relevant features for training the machine learning models. These techniques involve extracting and combining selected features to create an efficient feature set. Feature selection plays a vital role in achieving a good fit for machine-learning models as each feature has its significance for the target class. Therefore, an approach that incorporates only the features that contribute significantly to the final class prediction is developed. This approach offers several advantages such as easier interpretation of learning models,

reduction of model variances, and decreased training time and computational costs. To get the best feature solution, PCA is used as a feature selection approach in this study. By using PCA, the system's complexity is decreased while classification accuracy and stability are improved. The most useful features for the machine learning model can then be chosen by using PCA to find the principle components that capture the most important variances in the data. A detailed description of the PCA is given below.

## Principal component analysis

PCA is a commonly used technique for reducing the dimensionality of large datasets. This is accomplished by transforming a large set of features into a smaller set while retaining most of the relevant information from the original data. While reducing the number of features inherently sacrifices some accuracy, the key idea behind dimensionality reduction is to balance accuracy with simplicity. The dataset can be made easier to handle, explore, and visualize by simplifying it. Data processing is additionally sped up by machine learning algorithms, which can handle the data more effectively without a load of irrelevant features.

## Machine learning models

This study employed supervised machine learning classifiers to analyze maternal health risk data. The classifiers are implemented in Python using the 'Sci-kit learn' module. They are trained on a set of data samples dedicated for training purposes and evaluated on a separate test set that was unfamiliar to the classifiers. Several models, including RF, DT, ETC, LR, XGBoost, and SGD are individually utilized to construct the ensemble model. The optimal hyperparameter settings for these models are determined through a fine-tuning process. In this section, a brief overview of the ML classifiers used in this study is provided.

### Random forest

RF is a machine learning classifier that utilizes the combined effects of multiple decision trees trained on randomly selected subsets of the training data (*Breiman, 1996*; *Biau & Scornet, 2016*). The algorithm starts by splitting the initial training dataset into two distinct groups using a split function. This process is repeated until a termination condition is met, resulting in the creation of leaf nodes. The number of votes received by each leaf node determines the probability distribution associated with that node.

### Decision tree

DT is a simple machine learning technique that uses association rules to identify and predict target labels. It constructs a tree structure by selecting the root node and traversing it down to the leaf nodes for label prediction (*Manzoor et al., 2021*). Two primary methods used to determine the root node in a decision tree are the Gini index and information gain (IG). The IG criterion is commonly used as the default technique to select the top node in a decision tree.

### Logistic regression

LR is a statistical machine learning classifier that estimates the probability of mapping input features to discrete target variables using a sigmoid function (*Besharati, Naderan & Namjoo, 2019*; *Breiman, 1996*). The sigmoid function, represented by an S-shaped curve,

constrains the probability values for the discrete target variables. This makes LR particularly effective in classification problems. It is a powerful linear regression technique that can handle both linear and nonlinear datasets for classification and prediction tasks. LR is commonly used for binary data representation. The approach involves multiplying input values by weighted coefficients.

### Extreme gradient boosting

XGBoost is a classifier that functions similarly to gradient boosting but adds the ability to give each sample a weight, much like the AdaBoost classifier (*Ashraf et al., 2022*). The tree-based model XGBoost has become quite well-known recently. As opposed to gradient boosting, which trains weak learners (decision trees) sequentially, it trains several weak learners simultaneously. The increased speed of XGBoost is a result of this parallel training technique.

### Extra tree classifier

The ET consists of multiple de-correlated decision trees that are built using random subsets of training data features. The best feature is selected for each tree based on its Gini importance. ET employs averaging to reduce overfitting and improve prediction accuracy. What sets the ET classifier apart from other classifiers are two key differences. First, it does not bootstrap the data, meaning it samples without replacement. Second, nodes are randomly split rather than using the optimal split (*Muhammad Umer & Ashraf, 2022*).

### Stochastic gradient decent

The SGDC is an iterative method used to select the best smoothness characteristics for a differentiable or sub-differentiable objective function (*Umer et al., 2021*; *Majeed et al., 2021*). It is a stochastic approximation of gradient descent optimization, where the actual gradient computed from the complete dataset is replaced with an estimate obtained from a randomly selected subset of the data. SGDC is particularly effective in optimizing cost functions to determine optimal parameter and function coefficient values. It is a fast and efficient optimization technique that is commonly employed to learn linear classifiers with convex loss functions.

### Multilayer perceptron

An MLP has three layers: an input layer, an output layer, and one or more hidden layers. We undertook to fine-tune the experiment to produce the best prediction models, altering various parameters and investigating various layer counts (*Sarwat et al., 2022*). The following equation can be used to express a basic MLP model with one hidden layer as a function

$$h = g(W(1) \times x + b(1)) \tag{1}$$

$$y = s(W(2) \times h + b(2)) \tag{2}$$

In the above equation, $W(1)$ and $W(2)$ stand for the weight matrices, $b(1)$ and $b(2)$ for the bias vectors, $g$ for the hidden layer's activation function, and $s$ for the output layer's

activation function. The MLP's input is represented by $x$, the output of the hidden layer is represented by $h$, and the final output is represented by $y$. The MLP can be trained to learn from the input data and produce predictions by modifying the weights, biases, and activation functions.

### Convolutional neural network

CNN is a popular artificial neural network extensively used for various tasks (*Hameed et al., 2023*). It shares conceptual similarities with an MLP but differs in that each neuron in the CNN has its own activation function to map the weighted outputs. When an MLP incorporates multiple hidden layers, it is referred to as a deep MLP. The CNN's architecture allows it to exhibit invariance to translation and rotation. The CNN comprises three fundamental layers: a core layer, a pooling layer, and a fully connected layer, each with its own activation function.

## Proposed TreeNet model

The suggested model in this work combines two highly effective classifiers, the MLP, and ET classifiers, which are applied to the dataset for maternal health. The ET is an ensemble model based on trees, whereas the MLP is a neural model. These models are combined to create a strong hybrid model that takes advantage of both models. The reason for creating an ensemble of these two models is that they perform best among all other models individually. The soft voting criteria is used to combine the models, where the average probability for each class is determined by averaging the probabilities of each class predicted by each model. To arrive at a final prediction, this method considers the combined knowledge of the individual models. Figure 1 shows a graphical representation of the tree model, which helps to illustrate the ensemble model's structure and decision-making process.

The predictions of various machine learning algorithms are combined in the ensemble model to increase prediction accuracy and robustness. Both the MLP and ET models are independently trained on the same dataset for the ET+MLP ensemble model. Predicted probabilities are produced by each of these models for the various classes of the target variable. These projected probabilities are pooled to create a final forecast for each observation in the dataset. Taking a weighted average of the predicted probabilities is a typical technique for combining predictions. The weights allocated to each model's prediction are often decided based on how well they perform on a validation set or using methods like cross-validation. The ensemble model seeks to provide improved predictive performance and improve the overall accuracy and reliability of the forecasts by combining the advantages of the MLP, and ET models.

The proposed ensemble tree model uses the advantages of two different machine-learning methods to produce predictions that are more accurate. We can improve the model's capacity for generalization and reduce overfitting by training multiple models on the maternal health dataset and combining their predictions. The suggested ensemble model's operation is described by Algorithm 1.

---

**Algorithm 1** Ensemble of ETC and MLP.

---

**Input:** input data $(x, y)_{i=1}^{N}$

$M_{ETC}$ = Trained_ETC

$M_{MLP}$ = Trained_MLP

1: **for** $i = 1$ *to M* **do**

2:      **if** $M_{ETC} \neq 0$ & $M_{MLP} \neq 0$ & *training_set* $\neq 0$ **then**

3:          $ProbMLP - lowRisk = M_{MLP}.probability(lowRisk - class)$

4:          $ProbMLP - midRisk = M_{MLP}.probability(midRisk - class)$

5:          $ProbMLP - highRisk = M_{MLP}.probability(highRisk - class)$

6:          $ProbETC - lowRisk = M_{ETC}.probability(lowRisk - class)$

7:          $ProbETC - midRisk = M_{ETC}.probability(midRisk - class)$

8:          $ProbETC - highRisk = M_{ETC}.probability(highRisk - class)$

9:          Decision function $= max(\frac{1}{N_{classifier}} \sum_{classifier}$
         $(Avg_{(ProbETC-lowRisk, ProbMLP-lowRisk)}$
         $, (Avg_{(ProbETC-midRisk, ProbMLP-midRisk)}$
         $, (Avg_{(ProbETC-highRisk, ProbMLP-highRisk)}$

10:      **end if**

11:      Return final label $\widehat{p}$

12: **end for**

---

M: Total number of learning models in the voting (two in our case).

N: Total number of samples in the dataset for x-¿feature and y-¿target.

p-hat: Represents the predictive probabilities of each test sample.

n: Total test sample probabilities.

Both $\sum_{i}^{n} ETC_i$, $\sum_{i}^{n} MLP_i$ produce prediction probabilities for every test sample. After being aggregated for each test case, these probabilities are then subjected to the soft voting criterion, as shown in Fig. 2. The highest average probability among the classes is taken into account, and the projected probabilities from the two classifiers are combined, to determine the final class in the ensemble model. Using the class with the highest likelihood score as a starting point, the final prediction will be made.

$$\widehat{p} = argmax\{\sum_{i}^{n} ETC_i, \sum_{i}^{n} MLP_i\}. \tag{3}$$

To elucidate the capabilities of the proposed approach, let's consider an illustration. This approach entails passing a sample through the ETC and MLP components. Following this process, probability scores are assigned to each class. Specifically, for ETC, Class 1 (lowRisk), Class 2 (midRisk), and Class 3 (highRisk) have likelihood scores of 0.6, 0.7,

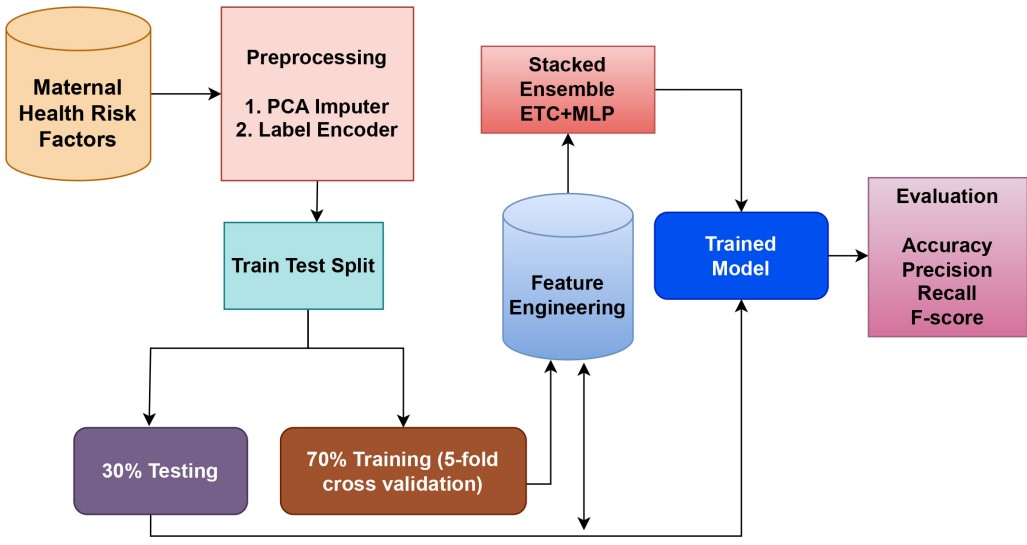

**Figure 1** **Proposed methodology workflow diagram.**

and 0.8, respectively. Similarly, for MLP, Class 1 (lowRisk), Class 2 (midRisk), and Class 3 (highRisk) have probability scores of 0.4, 0.5, and 0.6, respectively.

In this scenario, let $g(x)$ denote the probability score of $x$, where $x$ belongs to the three classes in the dataset. The domain of $x$ is confined to these three classes. Therefore, the probabilities for the three classes can be determined as follows:

P(lowRisk) = (0.6+0.4)/2 = 0.50
P(midRisk) = (0.7+0.5)/2 = 0.60
P(highRisk) = (0.8+0.6)/2 = 0.70

The final prediction will be highRisk, whose probability score is the largest, as shown below:

$$VC(ETC + MLP) = argmax(g(x)) \tag{4}$$

The final class is determined by the VC(ETC+MLP) using the highest average probability among the classes, and it combines the projected probabilities from both classifiers.

## Evaluation metrics

Several assessment criteria, such as accuracy, precision, recall, and F1 score are frequently employed to assess a model's performance. The true positive (TP), true negative (TN), false positive (FP), and false negative (FN) values from a confusion matrix can be used to determine these parameters.

Accuracy gauges how accurately the model's predictions are made overall and is computed as

$$Accuracy = \frac{TP + TN}{TP + TN + FP + FN} \tag{5}$$

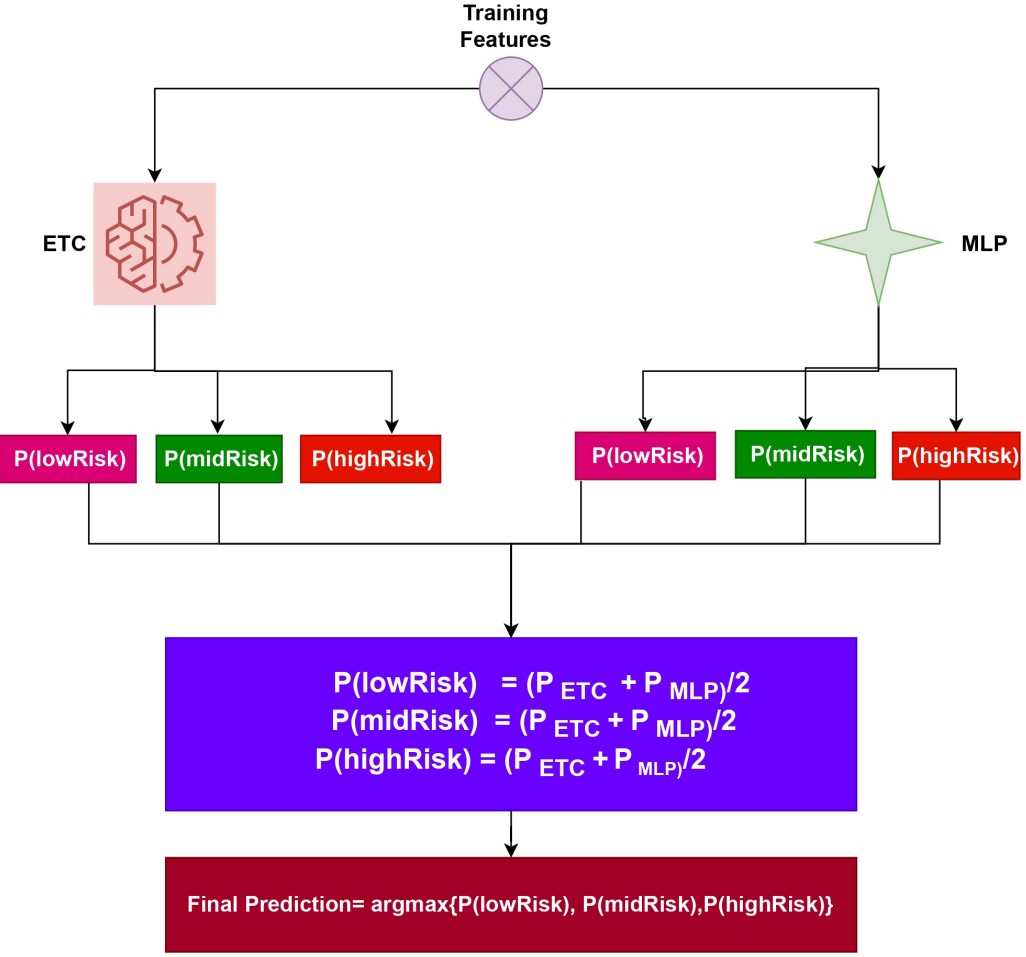

**Figure 2  Architecture of the proposed voting classifier.**

Precision measures how well the model can pick out positive cases from all those that are projected to be positive. It can be calculated using

$$Precision = \frac{TP}{TP + FP} \tag{6}$$

The recall is sometimes referred to as sensitivity or the true positive rate and measures the model's ability to properly detect positive events. The recall is determined by the formula

$$Recall = \frac{TP}{TP + FN} \tag{7}$$

F1 score balances the trade-offs between precision and recalls into a single parameter. The F1 score is determined as follows

$$F1\ score = 2 \times \frac{Precision \times Recall}{Precision + Recall} \tag{8}$$

It is the harmonic mean of precision and recall. These evaluation parameters provide valuable insights into the model's performance, considering different aspects such as

overall accuracy, precision in positive predictions, and the model's ability to detect positive instances.

# RESULTS AND DISCUSSION

The experimental results of various machine learning models are presented in this part from diverse angles. Both the original feature datasets and the datasets obtained through PCA are used in the performance evaluation of these models. The evaluation comprises determining how well ETC+MLP performs as feature extractors and classifiers. In terms of its ability to extract features, the performance of the proposed ETC+MLP technique is also compared with that of existing learning models.

## Experimental setup

This study conducted multiple experiments to evaluate and compare the performance of the proposed approach with various deep learning and machine learning models. All experiments are executed on a Windows 10 machine equipped with an Intel Core i7 7th generation processor. The proposed technique, as well as the machine learning and deep learning models, are implemented using Python frameworks such as TensorFlow, Keras, and Sci-kit Learn. The maternal health data is divided into 85% for training purposes and 15% for testing purposes. The experiments are conducted separately, using both the original feature set from the maternal health risk dataset and the feature set derived from PCA.

## Performance of models using original features

The initial set of experiments utilized the original feature set from the maternal health risk dataset. Table 3 presents the results obtained from various classifiers when applied to the original features. The outcomes indicate that the proposed ensemble model outperformed all individual learning models with an accuracy of 80.03%. The accuracy scores for the ETC and MLP classifiers are 77.08% and 79.45%, respectively. The deep learning model CNN achieved an accuracy score of 72.38%, while the tree-based model RF obtained the lowest accuracy among all models at 70.65%. It is important to note that the ensemble of tree classifiers with linear models (ETC+MLP) exhibited superior performance when applied to the original feature set.

When compared to linear models the tree ensemble model performs noticeably better. The efficacy of the voting model when handling a sizable number of features is the main driver of this development. The individual performances of the ETC and MLP classifiers are both satisfactory, and the combined results are even better. However, despite the commendable performance of the ensemble model the achieved accuracy still falls below the desired level for the accurate prediction of maternal health risks. Consequently, additional experiments are conducted to address this issue by utilizing PCA-extracted features.

## Performance of models using PCA features

Table 4 shows the results of the machine learning models developed using the dataset's PCA features. The outcomes of the subsequent set of experiments conducted using PCA

**Table 3** Results of the machine learning models obtained by using all features from the dataset.

| Model | Accuracy | Precision | Recall | F1 score |
|---|---|---|---|---|
| LR | 75.77 | 70.54 | 71.64 | 70.61 |
| DT | 72.24 | 70.51 | 70.45 | 70.27 |
| RF | 70.65 | 71.35 | 71.75 | 71.21 |
| SGD | 72.59 | 71.37 | 70.88 | 70.66 |
| ETC | 77.08 | 71.35 | 79.35 | 70.12 |
| XGBoost | 70.51 | 70.95 | 70.89 | 70.93 |
| MLP | 79.45 | 70.34 | 70.34 | 70.62 |
| CNN | 72.38 | 75.44 | 76.12 | 75.99 |
| Proposed | 80.03 | 82.46 | 82.21 | 82.33 |

**Table 4** Results of the machine learning models obtained by using PCA features from the dataset.

| Model | Accuracy | Precision | Recall | F1 score |
|---|---|---|---|---|
| LR | 88.33 | 88.76 | 90.46 | 89.59 |
| DT | 89.31 | 89.53 | 89.92 | 89.73 |
| RF | 90.31 | 90.84 | 90.73 | 90.77 |
| SGD | 90.92 | 89.71 | 90.42 | 90.19 |
| ETC | 93.42 | 92.76 | 93.12 | 92.91 |
| XGBoost | 91.52 | 92.42 | 93.43 | 92.78 |
| MLP | 95.43 | 96.68 | 97.50 | 97.05 |
| CNN | 90.73 | 91.76 | 91.43 | 91.68 |
| Proposed | 98.25 | 99.17 | 99.16 | 99.16 |

features to evaluate the effectiveness of both machine learning models and the proposed ensemble model indicate better results. The inclusion of PCA features aimed to select the most important features and enhance the accuracy of linear models. These PCA-extracted features are utilized for training and testing the machine learning models.

According to the experimental results, the proposed ensemble model outperforms all other models with a remarkable accuracy of 98.25%. When compared to the original features, this results in a considerable performance improvement of 18.22%. The performance of the individual linear models is also increased when PCA features are used. When compared to the original feature set MLP's accuracy is 95.43%, a 15.98% improvement while ETC's accuracy of 93.42% showed a 16.34% improvement. On the other hand, when using the PCA features, LR and the tree-based classifier DT obtained lower accuracy scores of 88.33% and 89.31%, respectively. When PCA is used for feature extraction, the models exhibit significantly better performance. Due to the strong connection between the features produced by PCA and the target class that makes the data linearly separable, linear models are better than other types of models.

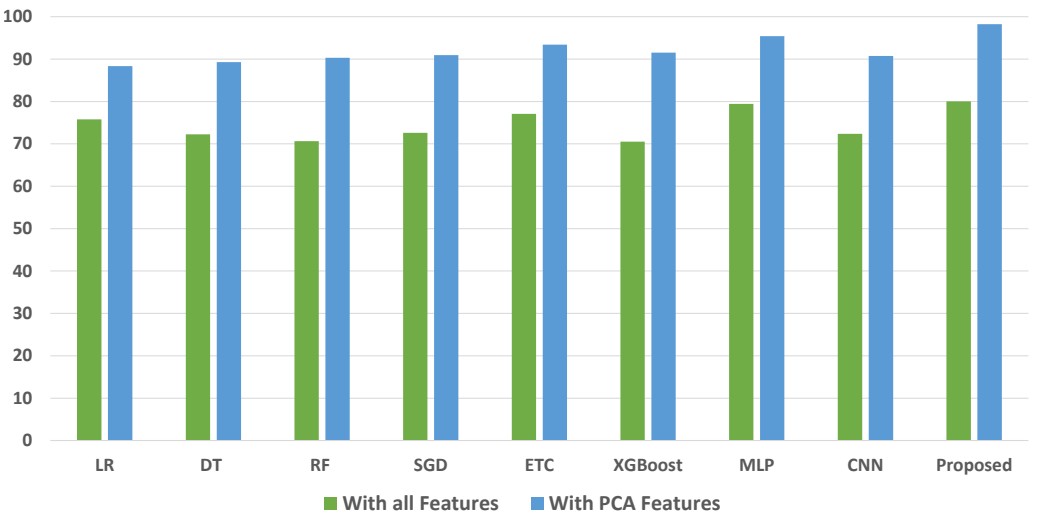

**Figure 3** Comparison of machine learning models in terms of accuracy.

## Comparison of machine learning models with original and PCA features

We conducted a thorough evaluation by comparing the performance of various machine learning models using both the original feature set and the features extracted through PCA. The objective is to assess the effectiveness of the proposed approach. The outcomes unequivocally demonstrated that incorporating PCA features in the second experiment, as opposed to utilizing the original dataset, led to a significant improvement in the performance of the machine learning models. To provide a comprehensive evaluation of their effectiveness, a comparison of machine learning models in terms of accuracy is presented in Fig. 3, in terms of precision in Fig. 4, in terms of recall in Fig. 5, and in terms of F1 score in Fig. 6.

## Results of the K-fold cross validation

We used K-fold cross-validation to further analyze the performance of the proposed approach. The results of 5-fold cross-validation are shown in Table 5, which demonstrates how well the proposed technique performs in terms of accuracy, precision, recall, and F1 score when compared to other models. Furthermore, it shows a low standard deviation, indicating stable performance throughout a variety of folds. These outcomes provide us with more assurance that the proposed technique is trustworthy and reliable.

## Discussion

The study conducted a thorough analysis of various machine learning models' performance using both the original features and features extracted through PCA. Initially, when applied to the original feature set, the proposed ensemble model outperformed individual models, achieving an accuracy of 80.03%. However, this fell short of the desired accuracy level for predicting maternal health risks.
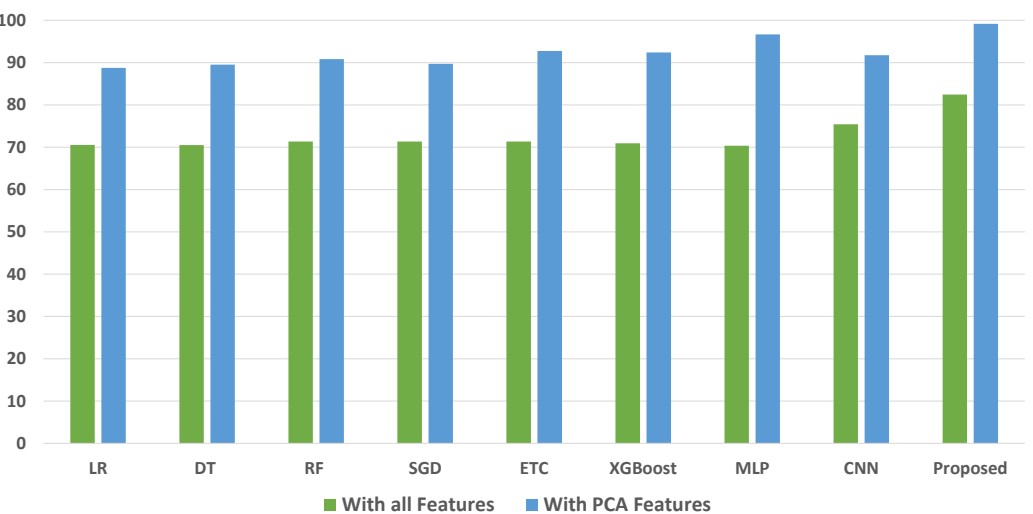

**Figure 4** **Comparison of machine learning models in terms of precision.**

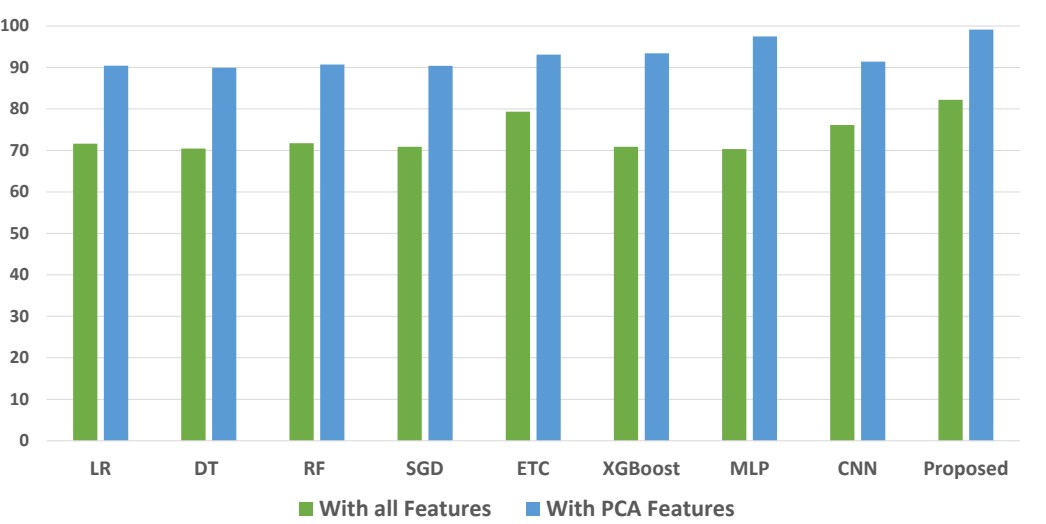

**Figure 5** **Comparison of machine learning models in terms of recall.**

The subsequent experiment utilizing PCA-extracted features showcased substantial improvements across models. The ensemble model's accuracy significantly increased to 98.25%, marking an 18.22% improvement over the original feature set.

Comparing models using both feature sets, the study highlighted the considerable enhancement in accuracy when employing PCA features. Linear models, especially, demonstrated substantial accuracy improvements, suggesting the features' ability to make the data more linearly separable.

Overall, the incorporation of PCA-extracted features notably boosted the predictive power of the models, particularly enhancing the ensemble model's performance, thus

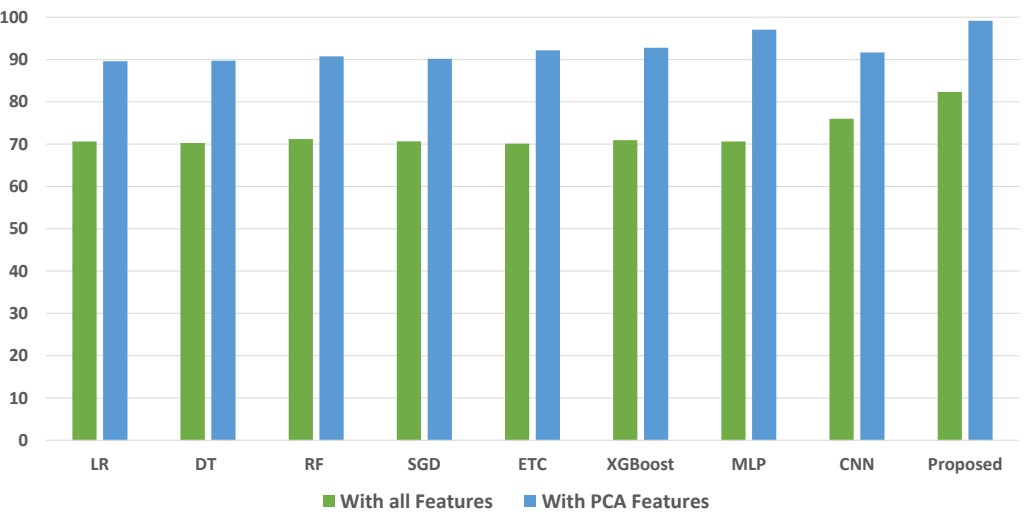

**Figure 6** Comparison of machine learning models in terms of F1 score.

**Table 5** Results of 5-fold cross-validation.

| Model | Accuracy | Precision | Recall | F1 score |
|---|---|---|---|---|
| 1st fold | 98.52 | 99.13 | 98.61 | 96.12 |
| 2nd fold | 98.25 | 98.34 | 98.74 | 96.23 |
| 3rd fold | 98.64 | 99.67 | 98.98 | 98.81 |
| 4th fold | 99.08 | 99.78 | 99.99 | 97.85 |
| 5th fold | 98.98 | 99.15 | 99.86 | 96.33 |
| **Average** | **98.89** | **99.52** | **98.49** | **98.11** |

Notes.
Values in bold indicate the average of all five-fold cross-validation results.

signifying the effectiveness of the approach in accurately predicting maternal health risks during pregnancy.

## Performance comparison with existing studies

A detailed comparison was made with nine pertinent research works that have produced models with an emphasis on accuracy improvement to assess the performance of the proposed model in comparison to current state-of-the-art models. These chosen works serve as comparisons for determining the efficiency of the proposed model and emphasizing its improvements over present methods. This research work offers insights into the superior performance of the proposed approach in terms of accuracy improvement by comparing the findings of the proposed model with those of the chosen state-of-the-art models. For instance, the SVM was used with a few selected features and attained an accuracy of 94% in *Irfan, Basuki & Azhar (2021)*. In Table 6, the proposed model and the current research on the same dataset are thoroughly compared in terms of performance. In *Mutlu et al. (2023)* and *Umoren, Silas & Ekong (2022)*, the authors applied DT and achieved 89.16% and 89.2% accuracy respectively. *Pawar et al. (2022)* have achieved 70.21% of

**Table 6  Performance comparison of the proposed approach with state-of-the-art models.**

| Reference | Proposed system | Achieved accuracy |
| --- | --- | --- |
| *Özsezer & Mermer (2021)* | Light GBM, CatBoost, | 88% |
| *Raza et al. (2022)* | SVM with DT-BiLTCN feature | 98% |
| *Mutlu et al. (2023)* | DT | 89.16% |
| *Umoren, Silas & Ekong (2022)* | DTCR | 89.2% |
| *Irfan, Basuki & Azhar (2021)* | SVM with selected features | 94% |
| *Pawar et al. (2022)* | RF | 70.21% |
| Proposed | VC(ETC+MLP) with PCA features | 98.25% |

accuracy. It can be observed that individual machine learning models have not shown good results for predicting maternal health because of the diversity of the dataset. In *Raza et al. (2022)*, the SVM model incorporating the DT-BiLTCN features attained a 98% accuracy, outperforming other models presented in Table 6. Their proposed DT-BiLTCN is based on complex learning model layers, especially with multiple layers and parameters, and could be challenging to interpret, making it harder to understand how and why certain predictions are made. However, the proposed model is a simple ensemble model with improved accuracy results. In terms of several performance evaluation parameters, this comparison reveals that the ensemble model using PCA features beats the other approaches.

## Shapley additive explanation

Understanding the relationships between inputs and outputs in machine or deep learning models can be challenging, given that these models are often perceived as opaque or black-box algorithms. This lack of transparency, particularly when working with labeled data, hampers a comprehensive comprehension of the importance of features in supervised learning on both a global and local scale. A recent advancement, the SHAP technique, addresses this issue by providing a quantitative approach to assess model interpretability. This breakthrough, initially introduced by *Lundberg & Lee (2017)* and subsequently expanded upon by *Lundberg et al. (2020)*, allows for a more nuanced understanding of the significance of elements within the model (*Ahmad, Eckert & Teredesai, 2018*; *Lundberg & Lee, 2017*).

SHAP employs the linear additive feature attribute method, drawing inspiration from cooperative game theory, to elucidate complex models. This method assigns an importance value to each attribute based on its impact on the model's predictions, contingent on the presence or absence of specific features during SHAP estimation. By employing this explanatory approach, the intricacies of complex models become more accessible through a simplified model. The application of the linear additive feature attribute technique, grounded in cooperative game theory principles, is extensively detailed in works by *Lundberg & Lee (2017)* and further expanded upon by *Lundberg et al. (2020)* (*Ahmad, Eckert & Teredesai, 2018*; *Lundberg & Lee, 2017*).

$$f(a) = g(a') = \phi_0 + \sum_{j=1}^{j} \phi_j a'_j \tag{9}$$

**Table 7** SHAPly maternal health feature importance table.

| Weight | Feature | Description |
|---|---|---|
| $0.1569 \pm 0.0438$ | Age | Age is a feature that represents the maturity of the body. |
| $0.0700 \pm 0.0478$ | Bs | Blood glucose level play a crucial role in the health of females |
| $0.0443 \pm 0.0458$ | Risk Level | Represents the intensity of risks associated with pregnancy. |
| $0.0210 \pm 0.0573$ | SystolicBP | Upper blood flow level. |
| $0.0087 \pm 0.0201$ | DiastolicBP% | Lower blood flow level. |
| $0.0087 \pm 0.0087$ | Heart Rate | Heart beat measured in beats per minute. |
| $0.0259 \pm 0.0234$ | Body Temperature | It will help to know about the symptoms related to the body temperature like fever, malaria, and other |

The original ensemble learning model under consideration is denoted as (a), while the simplified explanation model is represented as $g(a')$. Here, $a'_j$, where $j$ signifies a simplified input seismic attribute number, refers to these attributes. SHAP values, denoted as $\phi_j$, are calculated for all possible input orderings represented by $j$. The presence or absence of a specific seismic attribute is defined using an input vector, $a'_j$, during estimation. Finally, $\phi_0$ represents the model prediction when none of the attributes are considered during estimation. The comprehensive feature importance, calculated using SHAPly and arranged in descending order, is presented in Table 7. The analysis with SHAP highlights the significance of features in predicting maternal health. While SHAP feature importance surpasses traditional methods, relying solely on it offers only limited additional insights.

## CONCLUSIONS

Maternal health during pregnancy is of utmost importance as it directly impacts the well-being of both the mother and the developing fetus. While pregnancy is generally a natural and healthy process, it can also be associated with certain complications that require careful monitoring and management. This research work proposed a framework that consists of two portions for accurately diagnosing the risk related to maternal health. The first step is to extract significant features using the PCA feature engineering technique and the second part consists of the usage of the stacked ensemble voting classifier. The results with a high accuracy of 98.25% reveal that the proposed approach can perform superbly well for the early detection of risks related to maternal healthcare. The comparison with other state-of-the-art models also shows the superiority of the proposed model. The future work of this research work is to make a stacked ensembling of machine and deep learning models to further enhance the performance of the model on higher dimension datasets.

### Abbreviations

| | |
|---|---|
| **ANN** | Artificial Neural Network |
| **CNN** | Convolutional Neural Network |
| **LR** | Logistic Regression |

| | |
|---|---|
| **ETC** | Extra Tree Classifier |
| **VC** | Voting Classifier |
| **DT** | Decision Tree |
| **SGD** | Stochastic Gradient Descent |
| **RF** | Random Forest |
| **PCA** | Principle Component Analysis |
| **CPU** | Central Processing Unit |
| **GPU** | General Processing Unit |
| **OS** | Operating System |
| **RAM** | Random Processing Unit |
| **TN** | True Negative |
| **FN** | False Negative |
| **FP** | False Positive |
| **TP** | True Positive |

### Funding

This work was supported by the Princess Nourah bint Abdulrahman University Researchers Supporting Project number (PNURSP2024R440), Princess Nourah bint Abdulrahman University, Riyadh, Saudi Arabia. The funders had no role in study design, data collection and analysis, decision to publish, or preparation of the manuscript.

### Grant Disclosures

The following grant information was disclosed by the authors:
Princess Nourah bint Abdulrahman University, Riyadh, Saudi Arabia: PNURSP2024R440.

### Competing Interests

Imran Ashraf was an Academic Editor for PeerJ.

### Author Contributions

- Leila Jamel conceived and designed the experiments, performed the experiments, analyzed the data, prepared figures and/or tables, and approved the final draft.
- Muhammad Umer conceived and designed the experiments, performed the experiments, analyzed the data, performed the computation work, authored or reviewed drafts of the article, and approved the final draft.
- Oumaima Saidani conceived and designed the experiments, performed the experiments, prepared figures and/or tables, and approved the final draft.
- Bayan Alabduallah conceived and designed the experiments, performed the experiments, prepared figures and/or tables, and approved the final draft.
- Shtwai Alsubai performed the experiments, performed the computation work, prepared figures and/or tables, authored or reviewed drafts of the article, and approved the final draft.

- Farruh Ishmanov conceived and designed the experiments, performed the experiments, analyzed the data, performed the computation work, prepared figures and/or tables, authored or reviewed drafts of the article, and approved the final draft.
- Tai-hoon Kim conceived and designed the experiments, analyzed the data, prepared figures and/or tables, and approved the final draft.
- Imran Ashraf performed the experiments, analyzed the data, performed the computation work, authored or reviewed drafts of the article, and approved the final draft.

## Data Availability

The code and dataset are available at GitHub and Zenodo:

- https://github.com/MUmerSabir/MaternalHealthPeerJCS

- MUmerSabir. (2023). MUmerSabir/MaternalHealthPeerJCS: Maternal Health PeerJ CS code (MaternalHealth). Zenodo. https://doi.org/10.5281/zenodo.8114426.

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
