# Peer review of "Improving prediction of maternal health risks using PCA features and TreeNet model"

_PeerJ Computer Science, doi:10.7717/peerj-cs.1982_

## Round 0.1 · original submission · Major Revisions

According to the reviewer's comment, the quality of the paper has to be significantly removed by carefully addressing the technical and methodological issues. Hence, I would like to give my decision on Major revision. I strongly suggest the author(s) to carefully prepare a detail report on each questions raised by the reviewers and show the changes in the revised manuscript.

Reviewer 1 ·

Basic reporting

The basic reporting is normal.

Experimental design

The paper has presented an interesting problem, however, I have the below comments:
1. The diversity is a major problem and the ensemble classifier still struggles.
2. When the aim is to help analyzing the maternal health, the model should probably be explainable/interpretable, however, ensemble classifier would probably not help in this scenario.
3. It will probably be difficult to convince people to act on predictions when the method (ensemble one) is too complex.
4. Do the authors think if toxicity could too contribute to the maternal health? Please include the below papers while discussing this and shed some light:

5. Can the authors discuss of the performance of the method, when the input data is imbalanced? There are two paper that were published recently focusing on explainabilty of the model and data imbalancing. Please include these papers while discussing this point.

Validity of the findings

Pls check the above comments.

Reviewer 2 ·

Basic reporting

1. Last paragraph of introduction section do not have section number.
2. Related work is not about what others have done. Author need to include the key observation out of the related work along with possible future scope.

Experimental design

1. Dataset: The number of samples i the dataset is quite less(1014 only). How you justify applying deep learning on it.
2. PCA: is a standard method and description not required.
3. Similarly description of machine learning classifier seems to be redundant and need to be removed.

Validity of the findings

-A lot of have been reported literature, how do will you justify the novelty of this work.

Additional comments

Result and analysis section could be further improved.

·

Basic reporting

The paper is well-written in clear and professional English. However, there are a few issues that need attention:

1. The reference to WHO data from 2016 in line 57-58 seems outdated as it's now 2023. It is recommended using more recent data from WHO, which is expected to have annual updates. It is important to stay current with the latest information.

2. In line 94, it is suggested to use the common abbreviation "XGBoost" instead of "XGB" for "extreme gradient boosting" to make it more accessible to readers.

3. Table 2 could be improved by indicating the data types as "numerical" or "categorical" and providing additional information such as data range for "blood glucose level" and possible values for "intensity risk level" instead of using technical data types like "Int64," "Float64," or "object."

Experimental design

The experimental design and research methods are generally well-structured, but there are a couple of points that require attention:

4. In the section on "Machine Learning Models," the paper spends a significant amount of space introducing various machine learning models. While it's important to provide context for non-computer science readers, the extensive introduction of various machine learning models in this section might be excessive. Focus on the proposed model should be prioritized to maintain the paper's relevance to the chosen target journal.

5. The description of the proposed model/algorithm (lines 315-321) could benefit from improved clarity. While the pseudocode and flow chart offer a rough understanding, the text lacks precision. It's important to clarify terms like "M" in the pseudocode and the use of "p hat" as the "final label." In conventional machine learning, "p hat" usually represents estimated probabilities, and it's essential to maintain consistency. Please clarify the meaning of "N" and "n" in the context of equation 7, and whether "VC(ETC + MLP)" is a function or a variable. Additionally, explain the meaning of "g(x)" since it is not defined anywhere.

Validity of the findings

The paper presents a thorough experimental design, and the model achieves good results. However, there are areas where the validity of the findings can be improved:

6. For Tables 3 and 4, it is suggested to use "Acc," "Prec," "Rec," and "F1" as abbreviations instead of "A," "P," "R," and "F" for the sake of clarity. Also, avoid using "F" for the F1 score, as it can be confused with other F-scores such as F2, F3, etc.

7. In Table 5, the accuracy comparison between all models with and without PCA feature selection is presented. To improve the visual representation, consider using a bar plot to depict the comparison. This will make it easier for readers to discern the differences in accuracy, especially since accuracy results have already been shown in Tables 3 and 4.

8. In Table 7, the author compares the proposed system's accuracy with existing works. While the proposed system shows promising accuracy, it's essential to note that these numbers are directly copied from other papers, and the differences in datasets, features, and target classes could affect the meaning of these comparisons. It would be valuable to provide additional context or insights into these comparisons, acknowledging the potential disparities in the experimental setups.

Additional comments

The paper is generally well-written and easy to understand. The experiment design is thorough, and the proposed model achieves good results. However, the paper could benefit from providing more insights into why the model achieves good results rather than just presenting the results. An explanation of the underlying mechanisms, potential contributing factors, or novel approaches used in the research would add depth and context to the findings.

---

## Round 0.2 · accepted · Accept

Based on the positive comments from the reviewer, the paper will be accepted for possible publication in PeerJ Computer Science.

Reviewer 2 ·

Basic reporting

A lot has been improved in the revised version. It can be considered for publication.

Experimental design

As per the suggestion, the authors have incorporated the paper and thus enhanced the quality of the revised manuscript.

Validity of the findings

Even though the novelty of the work has not been justified as expected but the paper in the current form may be accepted for publication.